# Size-Dependent Critical Temperature and Anomalous Optical Dispersion in Ferromagnetic CrI_3_ Nanotubes

**DOI:** 10.3390/nano9020153

**Published:** 2019-01-26

**Authors:** Mohammed Moaied, Jisang Hong

**Affiliations:** 1Department of Physics, Faculty of Science, Zagazig University, 44519 Zagazig, Egypt; msmoayed@zu.edu.eg; 2Department of Physics, Pukyong National University, Busan 608-737, Korea

**Keywords:** first principles calculations, chromium(III) iodine, nanotubes, magnetic semiconductor, magnetic properties, optical properties

## Abstract

Using first principles calculations, we explored the magnetic and optical properties of chromium(III) iodide (CrI_3_) nanotubes (NTs) by changing their chirality and diameter. Here, we considered six types of NTs: (5,0), (5,5), (7,0), (10,0), (10,10), and (12,0) NTs. We found that both zigzag and armchair NTs had a ferromagnetic ground with a direct band gap, although the band gap was dependent on the chirality and diameter. Using the Monte Carlo simulation, we found that the Curie temperatures (T_c_) exhibited chirality and diameter dependence. In zigzag NTs, the larger the tube diameter, the larger the T_c_, while it decreased with increasing diameter in the armchair tube. We found that the T_c_ was almost doubled when the diameter increased two-fold. This finding may guide development of room temperature ferromagnetism in zigzag NTs. We also found that the CrI_3_ NTs displayed anisotropic optical properties and anomalous optical dispersion in the visible range. Specifically, the (10,0) zigzag NT had a large refractive index of 2 near the infrared region, while it became about 1.4 near blue light wavelengths. We also obtained large reflectivity in the ultraviolet region, which can be utilized for UV protection. Overall, we propose that the CrI_3_ NTs have multifunctional physical properties for spintronics and optical applications.

## 1. Introduction

Extensive studies have focused on two-dimensional (2D) materials because they display many peculiar physical properties not found in bulk or macroscopic structures. So far, numerous types of 2D materials have been fabricated. Most of these are non-magnetic, although few theoretical works have investigated the physical properties of magnetic 2D structures. Despite the difficulty of synthesizing intrinsic materials, it would be an interesting finding if a 2D ferromagnetic (FM) material with a finite band gap could be synthesized, because this would offer potential applications for spintronic or opto-spintronic devices. It was recently reported that the chromium(III) iodide (CrI_3_) monolayer has a finite band gap of 1.2 eV with an FM ground state. It was found that the CrI_3_ monolayer had a band gap of 1.2 eV with a critical Curie temperature of T_c_ = 45 K and strong magnetic anisotropy [1,2,3]. The pristine CrI_3_ layer consists of three monoatomic planes: one plane of chromium (Cr) atoms sandwiched between two atomic planes of iodine (I). Each Cr^3+^ ion is arranged in a honeycomb network of an edge-sharing octahedral coordinated by six I^−^ ions, and each I^−^ ion is bonded to two Cr ions. The isolated Cr atom has an electron configuration of 3d^5^4s^1^, and the largest atomic magnetic moment among all elements in the 3d transition metal series (6 µ_B_). Thus, the Cr in the pristine CrI_3_ layer is expected to have a +3 state with an electron configuration of 3d^3^4s^0^. This is consistent with the observed saturation magnetization of CrI_3_, which has a magnetic moment of around 3 µ_B_ per Cr atom [2].

After the discovery of carbon nanotubes (CNTs) in 1991 [4,5], both theoretical and experimental investigations have revealed that the unique structure of the CNT produces remarkable chiral dependent physical properties. This motivated the scientific community to search for another nanotube (NT) material with desirable features: boron nitride or phosphorene NT. From a structural point of view, a single-walled nanotube (SWNT) can be imagined as a cylinder with a diameter of only a few nanometers. The SWNTs may be rolled into one another to form ‘Russian dolls’ known as multi-walled carbon nanotubes (MWNTs) [4]. Unlike ordinary materials, it has been shown that the carbon based SWNT could be either metallic or semiconducting [6,7] according to its chirality. SWNTs can be constructed by rolling up the sheet within unit vectors (a_1_ and a_2_) along a certain direction C_h_ = na_1_ + ma_2_, also known as a chiral vector [8,9,10]. Chiral indices (n,m) are commonly used to label SWNTs. The chiral vector is perpendicular to a translational vector T pointing to the long axis of the SWNT. Both C_h_ and T vectors define the unit cell of a SWNT. Alternatively, one can use θ, the chiral angle between a_1_ and C_h_, to classify such materials. Due to the symmetry of the hexagonal lattice, |θ| ≤ 30°. SWNTs with chiral indices n = m (θ = 30°) or m = 0 (θ = 0°) have the highest symmetry; they are known as armchair (AC) or zigzag (ZZ) NTs, with a diameter, d, related to the chiral indices (m and n) by the equation $d=\frac{a}{\pi }\sqrt{n^{2}{+nm+m}^{2}}$.

Since CrI_3_ has a band gap with a magnetic state, it might be useful to explore the electron structure and magnetic and optical properties of the NTs by changing the chirality. Moreover, the critical temperature is one of the main issues in the magnetic materials, and the 2D CrI_3_ has a rather low Curie temperature of 45 K. Thus, it is necessary to increase the critical temperature for future device applications. So far, no studies on these issues in the CrI_3_ nanotube system are available. Thus, in our report, we aim to investigate the chirality and size-dependent magnetic state, band gap tuning, enhancement of critical temperature, and optical properties of the CrI_3_ NT, and propose that it can be utilized for potential multifunctional applications.

## 2. Numerical Method

We performed the ab initio simulations within the density functional theory [11,12] framework using the Vienna Ab initio Simulation Package (VASP) [13,14,15,16]. Exchange correlation interactions were treated with the generalized gradient approximation (GGA) within the Perdew–Burke–Ernzerhof (PBE) formulation [17,18]. The full-potential projected plane-wave framework [19,20] was used with an energy cutoff of 500 eV for the plane-wave basis set. We fully optimized the structure until the force on each atom was smaller than 0.01 eV Å^−1^ and the energy convergence reached up to 10^−5^ eV/atom, using the conjugate gradient method. For the one dimensional NTs, a unit cell with periodic boundary conditions was adopted to simulate the infinite *z* direction, and the vacuum distances of 15 Å along the *x* and *y* axes were applied to avoid interaction between two neighboring images. The Brillouin zone was sampled using the Monkhorst–Pack scheme using a 1 × 1 × 13 *k* point mesh for atomic and electronic relaxations. The electronic band structure was obtained from energy eigenvalues of 50 points along the Γ–X line in the Brillouin zone. Due to the localized 3D electrons of Cr, the correlation effect may alter the magnetic properties of CrI_3_ systems. It is therefore necessary to take the onsite Coulomb repulsion interaction of Cr 3D electrons into account, using the generalized gradient approximation and the Hubbard U term (GGA + U) method [21] to check the magnetic state of CrI_3_ NT systems with a moderate value of U_eff_ = 2.65 eV, which is adequate for obtaining a reasonable correction, as shown in the CrI_3_ monolayer system [22]. The optical properties were calculated from the frequency-dependent dielectric function *ε*(*ω*) = *ε*_1_(*ω*) + *iε*_2_(*ω*). The imaginary part of this function is determined by the following equation:(1)$$\varepsilon_{2}{}^{\alpha \beta }\left(\omega \right)=\;\frac{4\pi^{2}e^{2}}{\Omega }\underset{q\to 0}{lim}\frac{1}{q^{2}}\underset{c,v,k}{\sum }2\omega_{k}\delta \left(\epsilon_{ck}-\epsilon_{vk}-\omega \right)\times \langle u_{ck+e_{\alpha q}}|u_{vk}\rangle \langle u_{ck+e_{\beta }q}|u_{vk}\rangle {}^{*}$$ where *c* and *v* represent the conduction and valence band states, *k* is the wave vector, and *u_ck_* represents the wave function with a lattice constant periodicity. The real part is determined by the following equation:(2)$$\varepsilon_{1}{}^{\alpha \beta }\left(\omega \right)=1+\frac{2}{\pi }P\overset{\infty }{\underset{0}{\int }}\frac{\varepsilon_{2}^{\alpha \beta }\left(\omega^{\prime }\right)\omega^{\prime }}{\omega^{\prime 2}-\omega^{2}+i\eta }d\omega^{\prime }$$

The real and imaginary parts of the dielectric function are correlated by the well-known Kramers–Kronig relations [23,24].

## 3. Results and Discussion

### 3.1. Structural Characterization

Before we discuss the structure of the CrI_3_ SWNT, it is helpful to review the monolayer geometry. Figure 1a illustrates the atomic structure of a pristine CrI_3_ layer, which consists of three monoatomic planes X–M–X (X = I and M = Cr). All three planes belong to the same trigonal lattice with the basis vectors $\vec{a}$ and $\vec{b}$ of equal length, a = 6.8259 Å. The unit cell contains 6 I and 2 Cr atoms. The iodine planes are separated by distance δ = 2.88 Å, and the Cr atom is located at the center between two triangular iodine planes at a relative height of $\frac{\delta }{2}$. Similar to the single-walled CNTs, a CrI_3_ SWNT can be constructed by folding a monolayer into a tube form. The CrI_3_ SWNT with chiral index (n,m) is obtained by rolling up the helical vector in a given direction, where the chiral angle is also in the range $\theta =\left[0,\frac{\pi }{6}\right]\;$ [25,26] as presented in Figure 1a. Figure 1b,c displays two examples of typical zigzag (ZZ) and armchair (AC) CrI_3_ SWNTs. The CrI_3_ tube single wall consists of three coaxial cylinders (X–M–X) of thickness δ, and it endures different distortions when the monolayer is folded into a tube. Due to the advanced experimental technique, it is possible to fabricate CNTs with diameters of 1~2 nm [27]. Thus, in our work, we consider four types of NTs which have diameters of ~1–2 nm. Table 1 shows the structural information, such as lattice parameters (L), diameter (d), and Cr–I bond length in each tube in the ground state. The tube radius (d/2) corresponds to the distance between the axis of the tube and the cylinder of Cr atoms, and the interior and exterior iodine atoms are shrunken and stretched. A comparison of the calculated diameter $d_{\mathrm{calculated}}=\frac{a}{\pi }\sqrt{n^{2}+\mathrm{nm}+m^{2}}$ with the optimized value (d) yields an agreement.

To check the dynamic stability of CrI_3_ SWNT systems, we calculated the molecular dynamics, because it is a reliable tool to check whether a virtual structure is stable. The calculated dynamic stability and mechanical properties suggest that the 1D of CrI_3_ SWNT systems could be synthesized by appropriate experimental techniques on a suitable substrate, and that these materials can exist at ambient conditions. To show this, as an illustration, we present the snapshots of molecular dynamics at 5 ps with top and side views at 300 K for (5,0)-zigzag and (5,5)-armchair NTs in Appendix A.

### 3.2. Electronic Band Structures

We now present the electronic band structures and band gaps. Figure 2 and Figure 3 show the electronic band structures and the orbital projected partial density of states (PDOS). Note the overlap of the I-p_x_ and I-p_y_ states and the Cr-d_xz_ and d_yz_ states. This is due to the symmetry of the NT. It is well-known that CNTs can have either metallic or semiconducting band gaps according to the chirality, even if they have the same diameters [27]. However, all the CrI_3_ NTs had a direct band gap at the Γ-point, regardless of their chirality. We also found that these NTs displayed diameter- and chirality-dependent band gaps. For instance, in the ZZ CrI_3_ NT, both conduction band minimum (CBM) and valence band maximum (VBM) appeared in the majority spin state, and the band gaps increased from 0.435 (for (5,0) ZZ NT) to 0.909 eV (for (12,0) ZZ NT) with increasing diameter. As shown in the PDOS for the (5,0)-ZZ NT, we found strong hybridization between I-p and Cr-d states in the majority spin state. With increasing diameter, the repulsive interaction between Cr-d and I-p states in the majority spin state was further enhanced. In particular, the downshift of the I-p_z_ orbital in the majority spin valance band was clearly observed, and this resulted in an increase of the band gap. Additionally, both valence and conduction band edges appeared in the majority spin state. We also found that the I-p_z_ state contributed to the valence band edge, while the conduction band edge originated strongly hybridized states. However, the AC tube displayed different behavior. In the (5,5) configuration, both the CBM and VBM appeared in the majority spin state with a band gap of 0.695 eV, and the I-p_z_ orbital in the majority spin band formed the VBM edge. In contrast, for the (10,10) tube, the I-p_z_ orbital in the valence band with majority spin state was pushed down below the Fermi level. The I-p_(x,y)_ orbital in the minority spin band formed the VBM, and the I-Cr repulsion was suppressed. Consequently, we observed a reduced band gap of 0.427 eV. As described, the size-dependent band gap appeared differently in the zigzag and armchair NTs, and this can be understood from a structural point of view. For instance, the zigzag NTs will be identical to the monolayer when the tube diameter becomes infinite [27]. Thus, the increasing band gap of zigzag CrI_3_ NTs with diameter is reasonable. In contrast, the armchair NT showed a distorted structure, and the geometric feature was still far away from the 2D characteristic for the armchair NTs considered in our calculations. Thus, the band gap of armchair NTs showed a different tendency from that found in zigzag NTs. However, when the tube diameter becomes infinite, the armchair NT will be identical to the 2D monolayer. To show this, as an illustration, we present the structures of (10,0)-zigzag and (10,10)-armchair NTs in Appendix A. Despite the chirality- and diameter-dependent band gap, all the CrI_3_ NTs have a semiconductor band structure with a band gap. The robustness of semiconducting half-metallic materials will be advantageous for spintronics applications, because no extra process is required to sort which NTs have either metallic or semiconducting properties with the FM state.

### 3.3. Curie Temperature

We now explore the magnetic ground state. To determine the magnetic ground state, we considered three possible spin configurations: ferromagnetic (FM), nonmagnetic (NM), and antiferromagnetic (AFM) coupling between chromium atoms. In Appendix A, we present the energy difference between the FM, NM, and AFM for CrI_3_ SWNT systems. One of the most important physical properties of a ferromagnetic material is the Curie temperature (T_c_). Using Metropolis Monte Carlo (MC) simulations [28] based on the Ising model, we calculated the Curie temperature for CrI_3_ NTs (AC and ZZ). According to the Ising model, the Hamiltonian equation can be written as $\hat{H}=-\underset{i,j}{\sum }J{\hat{m}}_{i}\;{\hat{m}}_{j}$, where ${\hat{m}}_{i}\;$ and ${\hat{m}}_{j}$ are the magnetic moments (in µ_B_) at sites *i* and *j*, and *J* is the exchange parameter. For simplicity, only the nearest neighboring exchange interaction is taken into account, and $J=\frac{E_{ex}}{Nm^{2}}$ where *E_ex_*, *N*, and *m*^2^ represent the exchange energy defined by $E_{ex}=\left(E_{AFM}-E_{FM}\right)$, number of Cr atoms per unit cell, and square magnetic moments. In Table 2, we present calculated results. During the MC simulations, a 50 supercell was used to mimic the CrI_3_ NTs, and this was large enough to minimize the periodic constraints. The MC simulation allowed us to calculate variations in the average magnetic moment per unit cell at a given temperature. Figure 4 displays the temperature dependent magnetization curve. The magnetic moment retains a high spin state in the low temperature range, and then drops to near zero at the critical temperature.

To check the reliability of the MC calculations, we also calculated the Curie temperature of the 2D CrI_3_ layer to be 51 K. Note that the experimentally measured value was 45 K [1]. For the (5,0)-ZZ CrI_3_ NT, the Curie temperature (T_c_) was 33 K, while the T_c_ of (5,5)-AC CrI_3_ NT was increased almost twice when the diameter is doubled, and this T_c_ was slightly larger than that of the 2D layer. In (10,0)-ZZ NT, it was significantly enhanced to 86 K. Unlike the zigzag NTs, the Curie temperature of armchair CrI_3_-NTs decreased from 63 K to 48 K with increasing diameter. Indeed, the Curie temperature is closely related to the strength of exchange interaction, and we expect a higher T_c_ if the J is larger. As shown in Table 2, the J value of zigzag NT is increased in a bigger NT, while it is suppressed in armchair NTs. This can nicely account for the chiral- and size-dependent critical temperature found in our work. One of the main issues in the low dimensional magnetic materials is achieving a room temperature ferromagnetism. It is obvious that the T_c_ will finally converge to that of the 2D sheet when the diameter of ZZ NT becomes infinite. Nonetheless, as shown in Table 2, the larger ZZ NT shows a larger T_c_, and this suggests that the T_c_ will increase until the diameter reaches a certain critical value. Due to our limited computing power, we cannot fully investigate this issue with very big ZZ NT. As an alternative approach, we performed an estimation for the T_c_ based on rough approximation. After that, we considered (7,0)-ZZ and (12,0)-ZZ NTs. We first checked the diameter-dependent band gap of ZZ NTs. Appendix A presents the structural information and energy gap for (7,0)-ZZ NT and (12,0)-ZZ NT. The energy gaps (E_g_) for (7,0)-ZZ NT and (12,0)-ZZ NT are 0.702 and 0.909 eV. Based on our diameter dependent band gap, we extrapolated the band gap for (5,0)-ZZ, (7,0)-ZZ, (10,0)-ZZ, and (12,0)-ZZ NTs, and Appendix A shows the calculated result. By extrapolation, we find that the band gap of 2D layer will be about 1.30 eV, and this is in good agreement with the previously reported value of 1.2 eV [1]. Appendix A, and Appendix A shows the magnetic information for (7,0)-ZZ NT and (12,0)-ZZ NT. The calculated Curie temperatures were 71 K and 134 K for (7,0)-ZZ NT and (12,0)-ZZ NT, respectively. From the T_c_ obtained using the Monte Carlo simulation for (5,0)-ZZ, (7,0)-ZZ, (10,0)-ZZ, and (12,0)-ZZ NTs, we estimated the T_c_ by assuming a linear relationship. Figure 5 shows the calculated result. As mentioned above, the T_c_ should start to drop after a certain size of NT. Nonetheless, if we can assume that the T_c_ progresses linearly until the diameter reaches about 5 nm, then our calculation suggests that the room temperature ferromagnetic structure can be possible if the diameter reaches about ~5–6 nm. 

### 3.4. Optical Properties

We now discuss the optical properties. The most important physical quantity for optical properties is the frequency dependent dielectric function $\varepsilon \left(\omega \right)=\varepsilon_{1}\left(\omega \right)+i\varepsilon_{2}\left(\omega \right)$, because all the optical quantities can be extracted from this frequency dependent dielectric function. The dielectric function ε(ω) is dependent on the polarization of the incident light. In this report, we consider two types of incident light propagation: electromagnetic waves perpendicular (or parallel electric polarization to the tube axis E_||_) and parallel (or perpendicular electric polarization to the tube axis E_⊥_) to the tube axis. Figure 6a,b shows the real and imaginary parts of the frequency-dependent dielectric function ε(ω). It is clear, especially in the long wavelength case, that the ε(ω) strongly depends on the electric polarization and chirality of the NTs. Subsequently, we can expect the anisotropic optical properties to be more noticeable in long wavelength conditions. Since the $\varepsilon_{2}\left(\omega \right)$ is related to the optical transition by the incident light, it is important to understand this function. If we consider the conventional experimental situation, it will be more realistic to explore the electromagnetic wave propagation perpendicular to the tube axis. Thus, we focus on the parallel electric field polarization (E_||_) in the following discussion. From the calculated $\varepsilon_{2}\left(\omega \right)$, we found that the first peaks in the low energy region (between ~1.3–1.7 eV) were rather broad. Nonetheless, the essential characteristic can be understood from the band structure and DOS. Considering the optical transition selection rule, in the (5,0) and (5,5) NTs we found that the major contribution to the absorption occurred from I-p_z_ to Cr-d_z_^2^ transition, while transition from both I-p_z_ to Cr-d_z_^2^ and from I-p_x,y_ to Cr-d_xz,yz_ contributed to the absorption in (10,0) and (10,10) NTs. The second absorption peaks (between ~2.7–3.2 eV) were much broader than the first peaks. In the second peaks, in both zigzag and armchairs tubes, the transitions from Cr-d_yz,zx_ to I-p_x,y,_ and Cr-d_z_^2^ to I-p_z_ were responsible for the absorption. For optical device applications, both refractivity and reflectivity are key properties which can be extracted from the frequency-dependent dielectric function. Figure 5c presents the refractive index given by:(3)$$N\left(\omega \right)=\;\sqrt{\varepsilon_{1}\left(\omega \right)+i\varepsilon_{2}\left(\omega \right)}=n\left(\omega \right)+ik\left(\omega \right)$$ where *n*(*ω*) and *k*(*ω*) represent real and imaginary parts of the refractive index. The real part of *N*(*ω*) is called the index of refraction, and it is related to the speed of the electromagnetic wave, while the imaginary part is related to the wave absorption. We found that the refractive index is strongly sensitive to the wavelength and diameter of the NT, and such a dependency was clearly observed in long-wavelength cases. For instance, in the one edge of visible light (~380 nm = 3.3 eV), the (5,0)-ZZ NT (smallest diameter NT) had an *n*-value of 1.6, while the (10,10)-AC NT (largest diameter NT) had *n* = 1.30. By increasing the wavelength, we also found an interesting optical property. In many materials, the refractive index increases at shorter wavelengths, which is known as normal dispersion. For CrI_3_ NTs, however, the *n*(*ω*) becomes larger with increasing wavelength (anomalous dispersion). A large value of n was also observed near the infrared edge (~700–800 nm = ~1.8–1.6 eV). For instance, we obtained a refractive index of 2 for light propagating perpendicular to the tube axis (E_||_ polarization), and this value may have potential applications for infrared optical devices. As presented in Table 2, the (10,0)-ZZ tube had the highest critical temperature—almost twice that of the 2D sheet—as well as the largest refractive index. The refractive index varied from 1.4 to 2 in the visible range at different wavelengths. Thus, we find that the (10,0)-ZZ NT displays the most promising magnetic and optical properties for potential device applications. We also calculated the reflectivity given by:(4)$$R\left(\omega \right)={\left|\frac{1-\;\sqrt{\varepsilon_{1}\left(\omega \right)+i\varepsilon_{2}\left(\omega \right)}}{1+\;\sqrt{\varepsilon_{1}\left(\omega \right)+i\varepsilon_{2}\left(\omega \right)}}\right|}^{2}$$

Figure 5d shows the calculated results. For CrI_3_ NTs, both ZZ and AC tubes have a large reflectivity in the ultraviolet (UV) wavelength region, and this characteristic can be utilized for UV protection. On the other hand, the CrI_3_ NTs display rather weak reflectivity in the visible range for both directions of polarization, and all CrI_3_ NT systems would be optically transparent in a wide range of visible frequencies.

## 4. Conclusions

In summary, we investigated the geometry, electronic band structure, magnetic state, and optical properties of CrI_3_ SWNTs with different sets of chiral indices. Unlike the carbon nanotubes, we found that all CrI_3_ NT systems had a direct band gap with a ferromagnetic ground state, and the band gap was dependent on the chirality and the diameter. The band gap of zigzag NTs was increased with larger diameter, while the armchair NT showed the opposite trend. This can be understood from the geometric feature. We also calculated the Curie temperature using the Monte Carlo simulation, and the T_c_ displayed the same feature. In the zigzag NTs, the T_c_ was enhanced with increasing tube diameter, while the T_c_ of armchair NT decreased with increasing diameter. The (10,0) ZZ NT had a T_c_ of 86 K, which is almost twice of that found in the 2D CrI_3_ layer. We also explored the frequency dependent optical properties. We found that the optical properties were strongly anisotropic according to the electric polarization direction, and also dependent on the chirality and diameter of NTs. The anisotropic behavior was more noticeable in the long wavelength regime, and the anomalous optical dispersion was observed in the visible range because the refractive index was also increased in the long wavelength. In particular, the (10,0) ZZ NT, which had the largest Curie temperature in our systems, displayed a large refractive index of 2 near the infrared regime, while it became about 1.4 near the blue light wavelength. This wide variation may suggest that the ZZ NTs have potential infrared optical device applications. Additionally, the CrI_3_ NTs showed a large reflectivity near the ultraviolet regime, which can be used for UV protection. Overall, we propose that the CrI_3_ NTs have multifunctional physical properties for spintronics and optical applications.

## Figures and Tables

**Figure 1 nanomaterials-09-00153-f001:**
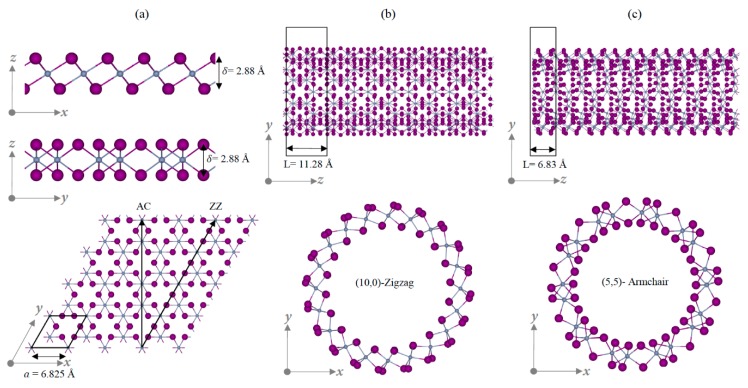
Atomic structure of chromium(III) iodide (CrI_3_): (**a**) side and top views of pristine 2D layer (**b**) illustration of (10,0) zizag (ZZ) nanotubes (NT) (**c**) (5,5) armchair (AC) NT with side and top views.

**Figure 2 nanomaterials-09-00153-f002:**
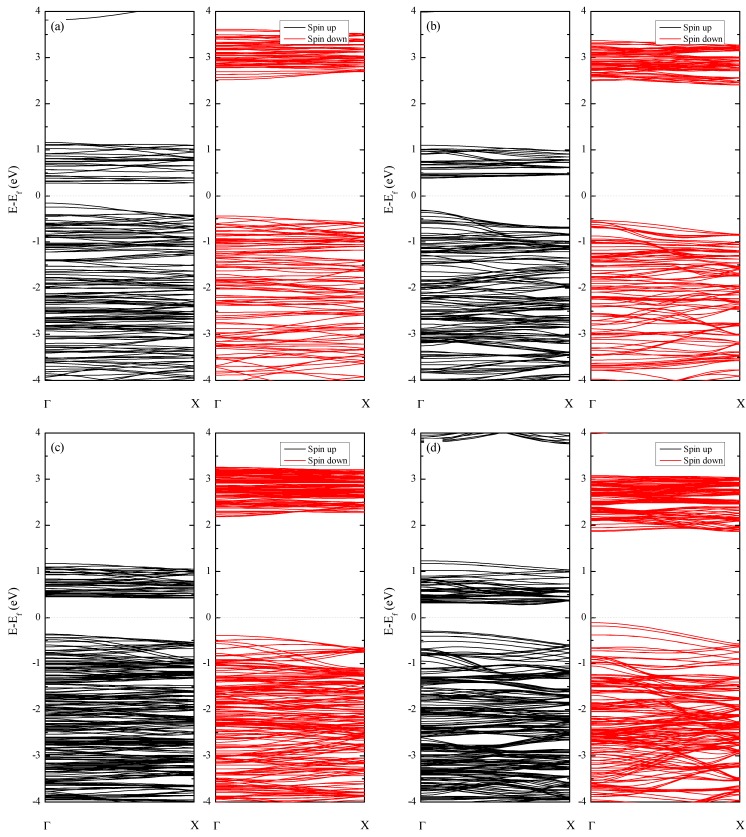
Electronic band structures of the: (**a**) (5,0) ZZ NT (**b**) (5,5) AC NT (**c**) (10,0) ZZ NT, and (**d**) (10,10) AC CrI_3_ NT.

**Figure 3 nanomaterials-09-00153-f003:**
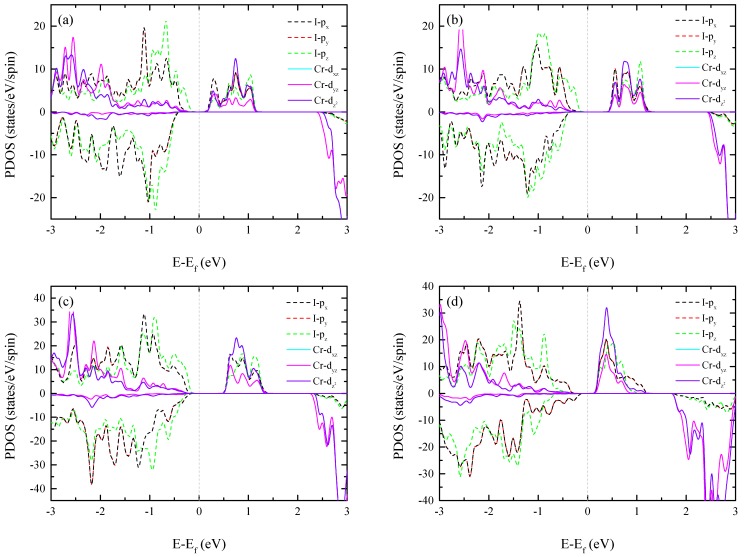
Orbital projected partial density of state (PDOS) of (**a**) (5,0) ZZ NT (**b**) (5,5) AC NT (**c**) (10,0) ZZ NT, and (**d**) (10,10) AC NT.

**Figure 4 nanomaterials-09-00153-f004:**
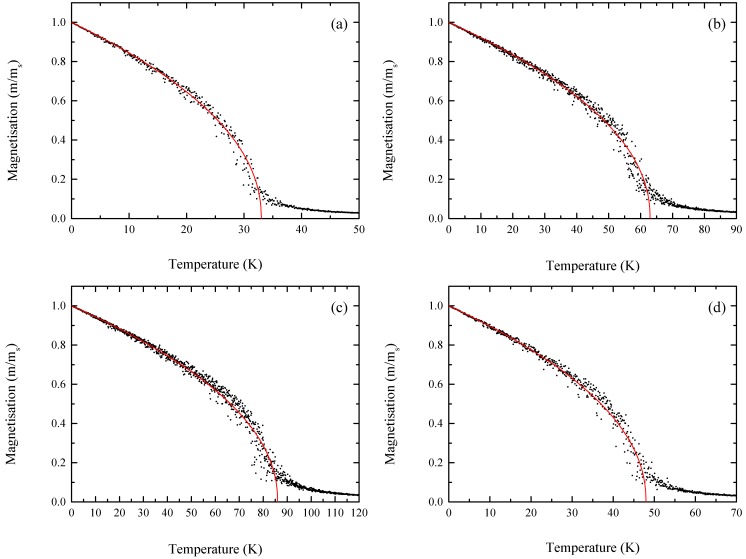
Temperature-dependent magnetization curve for (**a**) (5,0)-ZZ NT (**b**) (5,5)-AC NT (**c**) (10,0)-ZZ NT, and (**d**) (10,10)-AC NT.

**Figure 5 nanomaterials-09-00153-f005:**
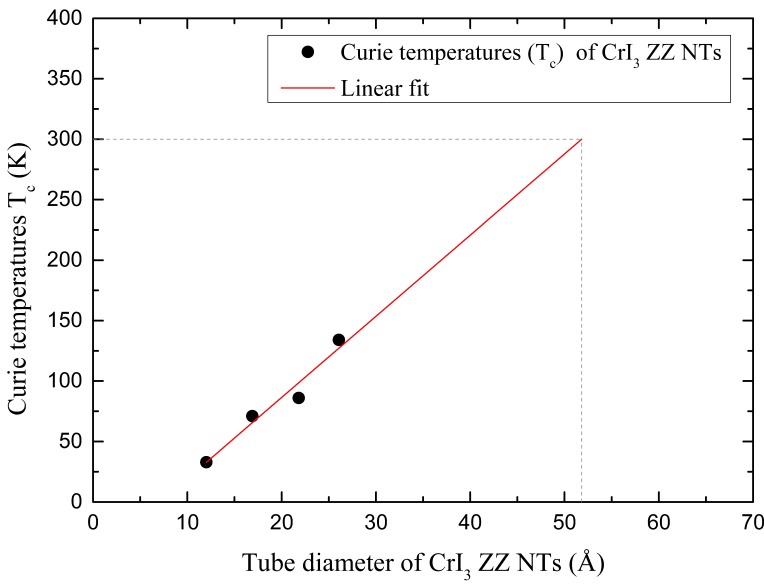
Diameter dependent Curie temperature of zigzag nanotubes.

**Figure 6 nanomaterials-09-00153-f006:**
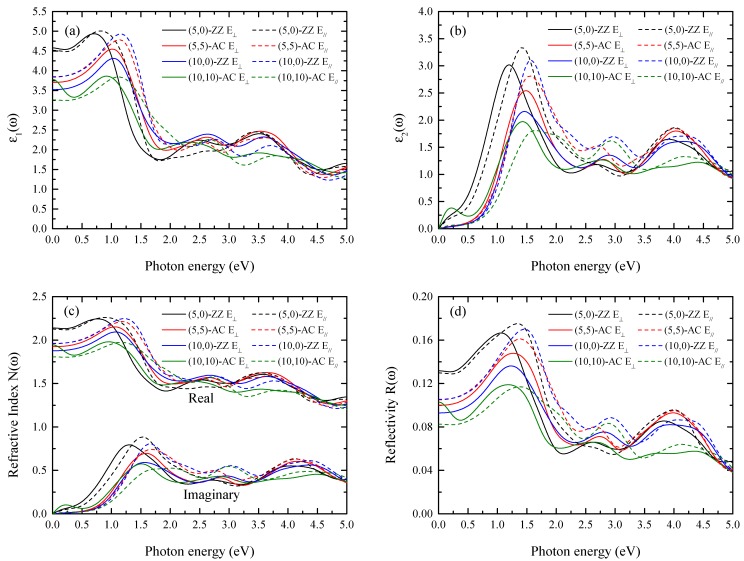
(**a**) Real and (**b**) imaginary parts of the frequency-dependent dielectric function, (**c**) real and imaginary parts of the refractive index, and (**d**) reflectivity for parallel (E_||_) and perpendicular (E_⊥_) electric field polarization with respect to the tube axis. Four tubes are presented: the (5,0)-ZZ, (5,5) AC, (10,0)-ZZ, and (10,10)-AC CrI_3_ SWNT.

**Table 1 nanomaterials-09-00153-t001:** Structural information of CrI_3_ NTs such as lattice parameters (L), the diameter (d), the calculated diameter (d_calculated_), Cr–I bond length (d_Cr–I_), the interlayer vertical distance between two iodine cylinders (δ), and energy gap (E_g_).

Compound	L (Å)	d (Å)	d_calculated_ (Å)	d_Cr–I_ (Å)	δ (Å)	E_g_ (eV)
(5,0)-ZZ	11.821	12.01	10.87	2.71	2.83	0.435
(5,5)-AC	6.825	19.30	18.82	2.68	2.93	0.695
(10,0)-ZZ	11.821	21.81	21.73	2.68	2.95	0.809
(10,10)-AC	6.825	35.93	37.65	2.69	2.96	0.427

**Table 2 nanomaterials-09-00153-t002:** Calculated local magnetic moments on the Cr site (M_Cr_), local magnetic moments on the I site (M_I_), total magnetic moments in a unit cell (M_cell_), exchange energy (E_ex_), exchange magnetic coupling (J), and Curie temperature (T_c_).

Compound	M_Cr_ (µ_B_)	M_I_ (µ_B_)	M_cell_ (µ_B_)	E_ex_ (×10^−3^ eV)	J (×10^−3^ eV)	T_c_ (K)
(5,0)-ZZ	3.55	−0.18	60.36	351.81	1.95	33
(5,5)-AC	3.47	−0.16	60.09	675.81	3.75	63
(10,0)-ZZ	3.42	−0.14	119.66	1859.97	5.17	86
(10,10)-AC	3.39	0.14	119.30	1044.06	2.90	48

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
