# Peer review of "Size-Dependent Critical Temperature and Anomalous Optical Dispersion in Ferromagnetic CrI3 Nanotubes"

_nanomaterials, 2019, doi:10.3390/nano9020153_

Reviewer 1 Report

Referee Report on paper “Size dependent critical temperature and anomalous optical dispersion in ferromagnetic CrI3 nanotubes” (nanomaterials-413131) by authors  Mohammed Moaied and JisangHong submitted to Nanomaterials

In present paper discussed the features of magnetic and optical properties for chromium(III) iodide (CrI3) nanotubes with different configurations. Data obtained using numerical calculations. This paper is interesting and contains useful information. The data are reliable and do not cause much doubt. Nevertheless, there are several sufficient points before the paper can be published. Please provide satisfied answers for my comments:

Is there any evidence that CrI3 exhibit ferromagnetic property? Not calculated. Really measured. Of course Cr3+ ion has 3mB. But for indirect exchange (in oxides or halcogenides) it is possible even antiferromagnetic or diamagnetic states. There are a lot examples when magnetic ion can not demonstrate ferromagnetic ordering with Tc.

Please provide information is this object can be produced in real.

Please explain why such low temperatures for magnetic ordering in this compound.

Author Response

Dear Editor and referee of Nanomaterials

First of all, I would like to thank you for handling our manuscript

Regarding the referee comments, I would like to provide my reply in the below 

Q) Is there any evidence that CrI3 exhibit ferromagnetic property? Not calculated. Really measured.

Ans) For the 2D CrI3 layer structure, it was very recently fabricated and published in Nature journal and it exhibits ferromagnetic property.

B. Huang, G. Clark, E. Navarro-Moratalla, D.R. Klein, R. Cheng, K.L. Seyler, D. Zhong, E. Schmidgall, M.A. McGuire, D.H. Cobden, W. Yao, D. Xiao, P. Jarillo-Herrero, X. Xu, Layer-dependent ferromagnetism in a van der Waals crystal down to the monolayer limit, Nature. 546 (2017) 270–273. doi:10.1038/nature22391.

Since this material (2D CrI3 structure) is very new, no experimental studies for nanotubes structures are available so far.

Of course Cr3+ ion has 3mB. But for indirect exchange (in oxides or halcogenides) it is possible even antiferromagnetic or diamagnetic states. There are a lot examples when magnetic ion can not demonstrate ferromagnetic ordering with Tc.

Ans) We agree with the referee in this point. So, we explored the magnetic ground state. To determine the magnetic ground state, we considered three possible spin configurations; ferromagnetic (FM), nonmagnetic (NM), and antiferromagnetic (AFM) coupling between chromium atoms. The table presents the energy difference between the FM, NM and AFM for CrI3 SWNT systems.

TABLE. Energy difference (in meV) between the FM, NM and AFM for (5,0)-ZZ, (5,5)-AC, (7,0)-ZZ, (10,0)-ZZ, (10,10)-AC, and (12,0)-ZZ CrI3 SWNT.

Compound

FM (meV)

NM (meV)

AFM (meV)

(5,0)-ZZ

0.00

438.66

351.81

(5,5)-AC

0.00

685.78

675.81

(7,0)-ZZ

0.00

1152.03

1082.74

(10,0)-ZZ

0.00

2175.82

1859.97

(10,10)-AC

0.00

1139.62

1044.06

(12,0)-ZZ

0.00

4475.62

3483.997

 Q) Please provide information is this object can be produced in real.

Ans) It is widely accepted that the first-principles calculations guide and assist the development of functional materials by high-precision calculations of electronic structure of materials. The accurate determination of electronic state is very important for understanding the mechanisms by which material functions come about so that their performance can be improved or the first principles calculations help synthesize new structure. There are many such examples. For instance:

Boron nitride nanotubes (BNNTs) are a polymorph of boron nitride. They were predicted in 1994 [1] and experimentally discovered in 1995 [2].

Phosphorene (BP) is predicted to be a strong competitor to graphene [3,4]. Phosphorene was first isolated in 2014 by mechanical exfoliation [5–7].

Chromium triiodide (CrI3) is predicted to be a ferromagnetic semiconductor (2D) material with many chromium trihalide CrX3 (X=F, Cl, Br, and I) [8,9]. Then was first isolated in 2017 [10].

We can still find many other evidences. In this regard, we used the first principles calculations and explored the physical properties of the CrI3 NTs. We believe that this work may stimulate to fabricate the nanotube structures and explore the various issues.

 [1]             A. Rubio, J.L. Corkill, M.L. Cohen, Theory of graphitic boron nitride nanotubes, Phys. Rev. B. 49 (1994) 5081–5084. doi:10.1103/PhysRevB.49.5081.

[2]             N.G. Chopra, R.J. Luyken, K. Cherrey, V.H. Crespi, M.L. Cohen, S.G. Louie, A. Zettl, Boron Nitride Nanotubes, Science. 269 (1995) 966–967. doi:10.1126/science.269.5226.966.

[3]             Five reasons phosphorene may be a new wonder material - MagLab, (n.d.). https://nationalmaglab.org/news-events/feature-stories/phosphorene-wonder-material (accessed May 15, 2018).

[4]             A. Carvalho, M. Wang, X. Zhu, A.S. Rodin, H. Su, A.H.C. Neto, Phosphorene: from theory to applications, Nat. Rev. Mater. 1 (2016) 16061. doi:10.1038/natrevmats.2016.61.

[5]             L. Li, Y. Yu, G.J. Ye, Q. Ge, X. Ou, H. Wu, D. Feng, X.H. Chen, Y. Zhang, Black phosphorus field-effect transistors, Nat. Nanotechnol. 9 (2014) 372–377. doi:10.1038/nnano.2014.35.

[6]             S.P. Koenig, R.A. Doganov, H. Schmidt, A.H. Castro Neto, B. Özyilmaz, Electric field effect in ultrathin black phosphorus, Appl. Phys. Lett. 104 (2014) 103106. doi:10.1063/1.4868132.

[7]             H. Liu, A.T. Neal, Z. Zhu, Z. Luo, X. Xu, D. Tománek, P.D. Ye, Phosphorene: An Unexplored 2D Semiconductor with a High Hole Mobility, ACS Nano. 8 (2014) 4033–4041. doi:10.1021/nn501226z.

[8]             W.-B. Zhang, Q. Qu, P. Zhu, C.-H. Lam, Robust intrinsic ferromagnetism and half semiconductivity in stable two-dimensional single-layer chromium trihalides, J. Mater. Chem. C. 3 (2015) 12457–12468. doi:10.1039/C5TC02840J.

[9]             J. Liu, Q. Sun, Y. Kawazoe, P. Jena, Exfoliating biocompatible ferromagnetic Cr-trihalide monolayers, Phys. Chem. Chem. Phys. 18 (2016) 8777–8784. doi:10.1039/C5CP04835D.

[10]          B. Huang, G. Clark, E. Navarro-Moratalla, D.R. Klein, R. Cheng, K.L. Seyler, D. Zhong, E. Schmidgall, M.A. McGuire, D.H. Cobden, W. Yao, D. Xiao, P. Jarillo-Herrero, X. Xu, Layer-dependent ferromagnetism in a van der Waals crystal down to the monolayer limit, Nature. 546 (2017) 270–273. doi:10.1038/nature22391.

 Q) Please explain why such low temperatures for magnetic ordering in this compound.

Ans) The Curie temperature (Tc) is closely related to the strength of the exchange interaction and we expect a low Tc if the J is small and this is already commented in the main text.

Reviewer 2 Report

The article is devoted to numerical simulations of chromium(III) iodide (CrI3) nanotubes in various geometry (diameter and chirality). Authors presented Monte Carlo simulation for obtaining the magnetic and optical properties. In my opinion, it is valuable theoretical work, and I suggest acceptation the article for print with some minor corrections. The article is well-written and present sufficient scientific value. I think, that it can be found very helpful for scientists focused on 2D structures. My minor remarks are listed below.

The authors claim, that nanotubes in zigzag conformation cannot be calculated due to limited computational power (page 8). I wonder, why the authors did not use some computational cluster since VASP software is available also for parallel computing? I understand, that using the rough model is justified in some cases, but much more valuable would be the calculations carried out with the use of the exact model. I do not ask for carrying out additional computations since it takes a lot of time, but my remark can be treated as advice for authors. The computational clusters are much more effective and convenient in comparison to personal computers.

Typos:

Line 8: first principles – should be first-principles

Line 10: sixtypes – should be six types

Line 47: structural – missing „a” before

Lines 98, 99, 105: the text seems to be not cleaned with some auxiliary marks (“??”)

Line 138: down shift 0 should be downshift

Line 147: size dependent – should be size-dependent

Line 203: estimation – missing „an” before

Line 219: at the right side of figure 5 some line is visible – probably some remain of the frame. The picture should be clean

Figure 4 caption: Temperature dependent – should be Temperature-dependent

Author Response

Dear Editor and referee of Nanomaterials

First of all, I would like to thank you for handling our manuscript

Regarding the referee comments, I would like to provide my reply in the below 

Q) The authors claim, that nanotubes in zigzag conformation cannot be calculated due to limited computational power (page 8). I wonder, why the authors did not use some computational cluster since VASP software is available also for parallel computing? I understand, that using the rough model is justified in some cases, but much more valuable would be the calculations carried out with the use of the exact model. I do not ask for carrying out additional computations since it takes a lot of time, but my remark can be treated as advice for authors. The computational clusters are much more effective and convenient in comparison to personal computers.

Ans) We have calculated these results using a computational clusters. But unfortunately, to make zigzag NT with a diameter around 5 nm, we have 448 atoms in a unit cell which is a very expensive calculation even with using many computational clusters. So in easy way, we extrapolated the Tc by assuming the linear relationship and proposed the diameter of the zigzag NT which has room temperature ferromagnetism.

Typos:

Line 8: first principles – should be first-principles

Ans) We have corrected this in the revised manuscript.

Line 10: sixtypes – should be six types

Ans) We have corrected this in the revised manuscript.

Line 47: structural – missing „a” before

Ans) We have corrected this in the revised manuscript.

Lines 98, 99, 105: the text seems to be not cleaned with some auxiliary marks (“??”)

Ans) We have corrected this in the revised manuscript.

Line 138: down shift 0 should be downshift

Ans) We have corrected this in the revised manuscript.

Line 147: size dependent – should be size-dependent

Ans) We have corrected this in the revised manuscript.

Line 203: estimation – missing „an” before

Ans) We have corrected this in the revised manuscript.

Line 219: at the right side of figure 5 some line is visible – probably some remain of the frame. The picture should be clean

Ans) We have corrected this in the revised manuscript.

Figure 4 caption: Temperature dependent – should be Temperature-dependent

Ans) We have corrected this in the revised manuscript.

Once again, we would like to thank referees for providing valuable comments.
